# Status-quo policy gradient in Multi-Agent Reinforcement Learning

## Abstract

Individual rationality, which involves maximizing expected individual return, does not always lead to optimal individual or group outcomes in multi-agent problems. For instance, in social dilemma situations, Reinforcement Learning (RL) agents trained to maximize individual rewards converge to mutual defection that is individually and socially sub-optimal. In contrast, humans evolve individual and socially optimal strategies in such social dilemmas. Inspired by ideas from human psychology that attribute this behavior in humans to the status-quo bias, we present a status-quo loss ($SQLoss$) and the corresponding policy gradient algorithm that incorporates this bias in an RL agent. We demonstrate that agents trained with $SQLoss$ evolve individually as well as socially optimal behavior in several social dilemma matrix games. To apply $SQLoss$ to games where cooperation and defection are determined by a sequence of non-trivial actions, we present $GameDistill$, an algorithm that reduces a multi-step game with visual input to a matrix game. We empirically show how agents trained with $SQLoss$ on a $GameDistill$ reduced version of the Coin Game evolve optimal policies.

## 1 Introduction

In sequential social dilemmas, individually rational behavior leads to outcomes that are sub-optimal for each individual in the group (Hardin, 1968; Ostrom, 1990; Ostrom et al., 1999; Dietz et al., 2003). Current state-of-the-art Multi-Agent Deep Reinforcement Learning (MARL) methods that train agents independently can lead to agents that play selfishly and do not converge to optimal policies, even in simple social dilemmas (Foerster et al., 2018; Lerer & Peysakhovich, 2017).

To illustrate why it is challenging to evolve optimal policies in such dilemmas, we consider the Coin Game (Foerster et al., 2018). Each agent can play either selfishly (pick all coins) or cooperatively (pick only coins of its color). Regardless of the other agent's behavior, the individually rational choice for an agent is to play selfishly, either to minimize losses (avoid being exploited) or to maximize gains (exploit the other agent). However, when both agents behave rationally, they try to pick all coins and achieve an average long term reward of $-0.5$. In contrast, if both play cooperatively, then the average long term reward for each agent is $0.5$. Therefore, when agents cooperate, they are both better off. Training Deep RL agents independently in the Coin Game using state-of-the-art methods leads to mutually harmful selfish behavior (Section 2.2).

The problem of how independently learning agents evolve optimal behavior in social dilemmas has been studied by researchers through human studies and simulation models (Fudenberg & Maskin, 1986; Green & Porter, 1984; Fudenberg et al., 1994; Kamada & Kominers, 2010; Abreu et al., 1990). A large body of work has looked at the mechanism of evolution of cooperation through reciprocal behaviour and indirect reciprocity (Trivers, 1971; Axelrod, 1984; Nowak & Sigmund, 1992; 1993; 1998), through variants of reinforcement using aspiration (Macy & Flache, 2002), attitude (Damer & Gini, 2008) or multi-agent reinforcement learning (Sandholm & Crites, 1996; Wunder et al., 2010), and under specific conditions (Banerjee & Sen, 2007) using different learning rates (de Cote et al., 2006) similar to WoLF (Bowling & Veloso, 2002) as well as using embedded emotion (Yu et al., 2015), social networks (Ohtsuki et al., 2006; Santos & Pacheco, 2006).

However, these approaches do not directly apply to Deep RL agents (Leibo et al., 2017). Recent work in this direction (Kleiman-Weiner et al., 2016; Julien et al., 2017; Peysakhovich & Lerer, 2018) focuses on letting agents learn strategies in multi-agent settings through interactions with

other agents. Leibo et al. (2017) defines the problem of social dilemmas in the Deep RL framework and analyzes the outcomes of a fruit-gathering game (Julien et al., 2017). They vary the abundance of resources and the cost of conflict in the fruit environment to generate degrees of cooperation between agents. Hughes et al. (2018) defines an intrinsic reward (inequality aversion) that attempts to reduce the difference in obtained rewards between agents. The agents are designed to have an aversion to both advantageous (guilt) and disadvantageous (unfairness) reward allocation. This handcrafting of loss with mutual fairness evolves cooperation, but it leaves the agent vulnerable to exploitation. LOLA (Foerster et al., 2018) uses opponent awareness to achieve high cooperation levels in the Coin Game and the Iterated Prisoner's Dilemma game. However, the LOLA agent assumes access to the other agent's network architecture, observations, and learning algorithms. This access level is analogous to getting complete access to the other agent's private information and therefore devising a strategy with full knowledge of how they are going to play. Wang et al. (2019) proposes an evolutionary Deep RL setup to evolve cooperation. They define an intrinsic reward that is based on features generated from the agent's past and future rewards, and this reward is shared with other agents. They use evolution to maximize the sum of rewards among the agents and thus evolve cooperative behavior. However, sharing rewards in this indirect way enforces cooperation rather than evolving it through independently learning agents.

Interestingly, humans evolve individual and socially optimal strategies in such social dilemmas without sharing rewards or having access to private information. Inspired by ideas from human psychology (Samuelson & Zeckhauser, 1988; Kahneman et al., 1991; Kahneman, 2011; Thaler & Sunstein, 2009) that attribute this behavior in humans to the status-quo bias (Guney & Richter, 2018), we present the $SQLoss$ and the corresponding status-quo policy gradient formulation for RL. Agents trained with $SQLoss$ evolve optimal policies in multi-agent social dilemmas without sharing rewards, gradients, or using a communication channel. Intuitively, $SQLoss$ encourages an agent to stick to the action taken previously, with the encouragement proportional to the reward received previously. Therefore, mutually cooperating agents stick to cooperation since the status-quo yields higher individual reward, while unilateral defection by any agent leads to the other agent also switching to defection due to the status-quo loss. Subsequently, the short-term reward of exploitation is overcome by the long-term cost of mutual defection, and agents gradually switch to cooperation.

To apply $SQLoss$ to games where a sequence of non-trivial actions determines cooperation and defection, we present $GameDistill$, an algorithm that reduces a dynamic game with visual input to a matrix game. $GameDistill$ uses self-supervision and clustering to extract distinct policies from a sequential social dilemma game automatically.

Our key contributions can be summarised as:

1. We introduce a **Status-Quo** loss ($SQLoss$, Section 2.3) and an associated policy gradient-based algorithm to evolve optimal behavior for agents playing matrix games that can act in either a cooperative or a selfish manner, by choosing between a cooperative and selfish policy. We empirically demonstrate that agents trained with the $SQLoss$ evolve optimal behavior in several social dilemmas iterated matrix games (Section 4).

2. We propose $GameDistill$ (Section 2.4), an algorithm that reduces a social dilemma game with visual observations to an iterated matrix game by extracting policies that implement cooperative and selfish behavior. We empirically demonstrate that $GameDistill$ extracts cooperative and selfish policies for the Coin Game (Section 4.2).

3. We demonstrate that when agents run $GameDistill$ followed by MARL game-play using $SQLoss$, they converge to individually as well as socially desirable cooperative behavior in a social dilemma game with visual observations (Section 4.2).

## 2 APPROACH

### 2.1 SOCIAL DILEMMAS MODELED AS ITERATED MATRIX GAMES

To remain consistent with previous work, we adopt the notations from Foerster et al. (2018). We model social dilemmas as general-sum Markov (simultaneous move) games. A multi-agent Markov game is specified by $G = \langle S, A, U, P, r, n, \gamma \rangle$. $S$ denotes the state space of the game. $n$ denotes the

number of agents playing the game. At each step of the game, each agent $a \in A$, selects an action $u^a \in U$. $\vec{u}$ denotes the joint action vector that represents the simultaneous actions of all agents. The joint action $\vec{u}$ changes the state of the game from $s$ to $s'$ according to the state transition function $P(s'|\vec{u}, s) : S \times \mathbf{U} \times S \rightarrow [0, 1]$. At the end of each step, each agent $a$ gets a reward according to the reward function $r^a(s, \vec{u}) : S \times \mathbf{U} \rightarrow \mathbb{R}$. The reward obtained by an agent at each step is a function of the actions played by all agents. For an agent $a$, the discounted future return from time $t$ is defined as $R_t^a = \sum_{l=0}^{\infty} \gamma^l r_{t+l}^a$, where $\gamma \in [0, 1)$ is the discount factor. Each agent independently attempts to maximize its expected discounted return.

Matrix games are the special case of two-player perfectly observable Markov games (Foerster et al., 2018). Table 1 shows examples of matrix games that represent social dilemmas. Consider the Prisoner's Dilemma game in Table 1a. Each agent can either cooperate ($C$) or defect ($D$). Playing $D$ is the rational choice for an agent, regardless of whether the other agent plays $C$ or $D$. Therefore, if both agents play rationally, they each receive a reward of $-2$. However, if each agent plays $C$, then it will obtain a reward of $-1$. This fact that individually rational behavior leads to a sub-optimal group (and individual) outcome highlights the dilemma.

In Infinitely Iterated Matrix Games, agents repeatedly play a particular matrix game against each other. In each iteration of the game, each agent has access to the actions played by both agents in the previous iteration. Therefore, the state input to an RL agent consists of both agents' actions in the previous iteration of the game. We adopt this state formulation as is typically done in such games (Press & Dyson, 2012; Foerster et al., 2018). The infinitely iterated variations of the matrix games in Table 1 represent sequential social dilemmas. We refer to infinitely iterated matrix games as iterated matrix games in subsequent sections for ease of presentation.

## 2.2 Learning Policies in Iterated Matrix Games: The Selfish Learner

The standard method to model agents in iterated matrix games is to model each agent as an RL agent that independently attempts to maximize its expected total discounted reward. Several approaches to model agents in this way use policy gradient-based methods (Sutton et al., 2000; Williams, 1992). Policy gradient methods update an agent's policy, parameterized by $\theta^a$, by performing gradient ascent on the expected total discounted reward $\mathbb{E}[R_0^a]$. Formally, let $\theta^a$ denote the parameterized version of an agent's policy $\pi^a$ and $V_{\theta^1, \theta^2}^a$ denote the total expected discounted reward for agent $a$. Here, $V^a$ is a function of the policy parameters $(\theta^1, \theta^2)$ of both agents. In the $i^{th}$ iteration of the game, each agent updates $\theta_i^a$ to $\theta_{i+1}^a$, such that it maximizes it's total expected discounted reward. $\theta_{i+1}^a$ is computed as follows:

$$\theta_{i+1}^1 = argmax_{\theta^1} V^1(\theta^1, \theta_i^2) \quad \text{and} \quad \theta_{i+1}^2 = argmax_{\theta^2} V^2(\theta_i^1, \theta^2) \tag{1}$$

For agents trained using reinforcement learning, the gradient ascent rule to update $\theta_{i+1}^1$ is,

$$f_{nl}^1 = \nabla_{\theta_1^i} V^1(\theta_i^1, \theta_i^2) \cdot \delta \quad \text{and} \quad \theta_{i+1}^1 = \theta_i^1 + f_{nl}^1(\theta_i^1, \theta_i^2) \tag{2}$$

where $\delta$ is the step size of the updates. In the Iterated Prisoner's Dilemma (IPD) game, agents trained with the policy gradient update method converge to a sub-optimal mutual defection equilibrium (Figure 3a, Lerer & Peysakhovich (2017)). This sub-optimal equilibrium attained by Selfish Learners motivates us to explore alternative methods that could lead to a desirable cooperative equilibrium. We denote the agent trained using policy gradient updates as a Selfish Learner ($SL$).

## 2.3 Learning Policies in Iterated Matrix Games: The Status-Quo Aware Learner ($SQLoss$)

Figure 1 shows the high-level architecture of our approach.

### 2.3.1 $SQLoss$: Intuition

Why do independent, selfish learners converge to mutually harmful behavior in the IPD? To understand this, consider the payoff matrix for a single iteration of the IPD in Table 1a. In each iteration, an agent can play either $C$ or $D$. Mutual defection ($DD$) is worse for each agent than mutual cooperation ($CC$). However, one-sided exploitation ($DC$ or $CD$) is better than mutual cooperation for the exploiter and far worse for the exploited. Therefore, as long as an agent perceives the possibility

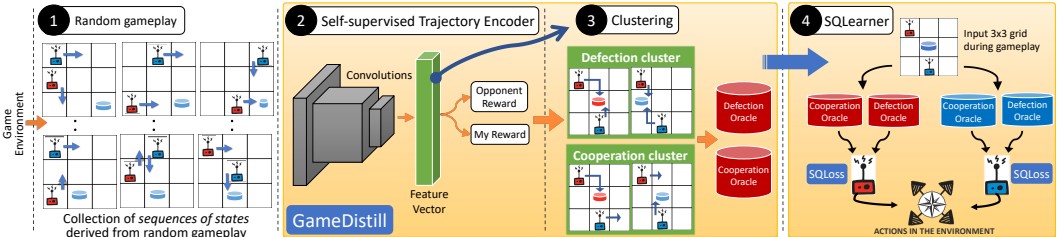

Figure 1: High-level architecture of our approach. Each agent runs $GameDistill$ by performing steps $(1), (2), (3)$ individually to obtain two oracles per agent. During game-play(4), each agent (with $SQLoss$) takes either the action suggested by the cooperation or the defection oracle

of exploitation, it is drawn to defect, both to maximize the reward (through exploitation) and minimize its loss (through being exploited). To increase the likelihood of cooperation, it is important to reduce instances of exploitation between agents. We posit that, if agents either mutually cooperate $(CC)$ or mutually defect $(DD)$, then they will learn to prefer $C$ over $D$ and achieve a socially desirable equilibrium. (for a detailed illustration of the evolution of cooperation, see Appendix C, which is part of the Supplementary Material)

Motivated by ideas from human psychology that attribute cooperation in humans to the status-quo bias (Guney & Richter, 2018), we introduce a status-quo loss ($SQLoss$) for each agent, derived from the idea of imaginary game-play (Figure 2). Intuitively, the loss encourages an agent to imagine an episode where the status-quo (current situation) is repeated for several steps. This imagined episode causes the exploited agent (in $DC$) to perceive a continued risk of exploitation and, therefore, quickly move to $(DD)$. Hence, for the exploiting agent, the short-term gain from exploitation $(DC)$ is overcome by the long-term loss from mutual defection $(DD)$. Therefore, agents move towards mutual cooperation $(CC)$ or mutual defection $(DD)$. With exploitation (and subsequently, the fear of being exploited) being discouraged, agents move towards cooperation.

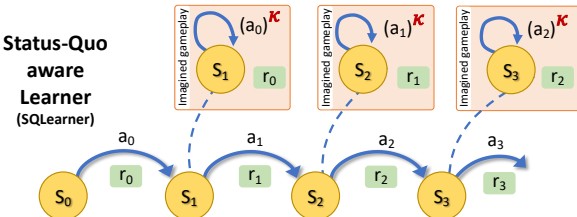

Figure 2: Intuition behind $Status-Quo$-aware learner. At each step, the $SQLoss$ encourages an agent to imagine the consequences of sticking to the status-quo by imagining an episode where the status-quo is repeated for $\kappa$ steps. Section 2.3 describes $SQLoss$ in more detail.

### 2.3.2 $SQLoss$: FORMULATION

We describe below the formulation of SQLoss with respect to agent 1. The formulation for agent 2 is identical to that of agent 1. Let $\tau_a = (s_0, u_0^1, u_0^2, r_0^1, \cdots s_T, u_T^1, u_T^2, r_T^1)$ denote the collection of an agent's experiences after $T$ time steps. Let $R_t^1(\tau_1) = \sum_{l=t}^{T} \gamma^{l-t} r_l^1$ denote the discounted future return for agent 1 starting at $s_t$ in actual game-play. Let $\hat{\tau}_1$ denote the collection of an agent's **imagined** experiences. For a state $s_t$, where $t \in [0, T]$, an agent imagines an episode by starting at $s_t$ and repeating $u_{t-1}^1, u_{t-1}^2$ for $\kappa_t$ steps. This is equivalent to imagining a $\kappa_t$ step repetition of already played actions. We sample $\kappa_t$ from a Discrete Uniform distribution $\mathbb{U}\{1, z\}$ where $z$ is a hyper-parameter $\geq 1$. To simplify notation, let $\phi_t(s_t, \kappa_t)$ denote the ordered set of state, actions, and rewards starting at time $t$ and repeated $\kappa_t$ times for imagined game-play. Let $\hat{R}_t^1(\hat{\tau}_1)$ denote the discounted future return starting at $s_t$ in imagined status-quo game-play.

$$\phi_t(s_t, \kappa_t) = \left[ (s_t, u_{t-1}^1, u_{t-1}^2, r_{t-1}^1)_0, \ (s_t, u_{t-1}^1, u_{t-1}^2, r_{t-1}^1)_1, \ \cdots, (s_t, u_{t-1}^1, u_{t-1}^2, r_{t-1}^1)_{\kappa_t - 1} \right] \tag{3}$$

$$\hat{\tau}_1 = \big(\phi_t(s_t, \kappa_t),\ (s_{t+1}, u^1_{t+1}, u^2_{t+1}, r^1_{t+1})_{\kappa_t+1},\ \cdots,\ (s_T, u^1_T, u^2_T, r^1_T)_{T+\kappa_t-t}\big) \tag{4}$$

$$\hat{R}^1_t(\hat{\tau}_1) = \Big(\frac{1-\gamma^\kappa}{1-\gamma}\Big)r^1_{t-1} + \gamma^\kappa R^1_t(\tau_1) = \Big(\frac{1-\gamma^\kappa}{1-\gamma}\Big)r^1_{t-1} + \gamma^\kappa \sum_{l=t}^{T}\gamma^{l-t}r^1_l \tag{5}$$

$V^1_{\theta^1,\theta^2}$ and $\hat{V}^1_{\theta^1,\theta^2}$ are approximated by $\mathbb{E}[R^1_0(\tau_1)]$ and $\mathbb{E}[\hat{R}^1_0(\hat{\tau}_1)]$ respectively. These $V$ values are the expected rewards conditioned on both agents' policies $(\pi^1, \pi^2)$. For agent 1, the regular gradients and the Status-Quo gradients, $\nabla_{\theta^1}\mathbb{E}[R^1_0(\tau_1)]$ and $\nabla_{\theta^1}\mathbb{E}[\hat{R}^1_0(\hat{\tau}_1)]$, can be derived from the policy gradient formulation as

$$\nabla_{\theta^1}\mathbb{E}[R^1_0(\tau_1)] = \mathbb{E}[R^1_0(\tau_1)\nabla_{\theta^1}log\pi^1(\tau_1)] = \mathbb{E}\Big[\sum_{t=1}^{T}\nabla_{\theta^1}log\pi^1(u^1_t|s_t)\cdot\sum_{l=t}^{T}\gamma^l r^1_l\Big]$$

$$= \mathbb{E}\Big[\sum_{t=1}^{T}\nabla_{\theta^1}log\pi^1(u^1_t|s_t)\gamma^t\big(R^1_t(\tau_1) - b(s_t)\big)\Big] \tag{6}$$

$$\nabla_{\theta^1}\mathbb{E}[\hat{R}^1_0(\hat{\tau}_1)] = \mathbb{E}\,[\hat{R}^1_0(\hat{\tau}_1)\nabla_{\theta^1}log\pi^1(\hat{\tau}_1)]$$

$$= \mathbb{E}\Big[\sum_{t=1}^{T}\nabla_{\theta^1}log\pi^1(u^1_{t-1}|s_t)\times\Big(\sum_{l=t}^{t+\kappa}\gamma^l r^1_{t-1} + \sum_{l=t}^{T}\gamma^{l+\kappa}r^1_l\Big)\Big]$$

$$= \mathbb{E}\Big[\sum_{t=1}^{T}\nabla_{\theta^1}log\pi^1(u^1_{t-1}|s_t)\times\Big(\Big(\frac{1-\gamma^\kappa}{1-\gamma}\Big)\gamma^t r^1_{t-1} + \gamma^\kappa\sum_{l=t}^{T}\gamma^l r^1_l\Big)\Big] \tag{7}$$

$$= \mathbb{E}\Big[\sum_{t=1}^{T}\nabla_{\theta^1}log\pi^1(u^1_{t-1}|s_t)\gamma^t\big(\hat{R}^1_t(\hat{\tau}_1) - b(s_t)\big)\Big]$$

where $b(s_t)$ is a baseline for variance reduction.

Then the update rule $f_{sql,pg}$ for the policy gradient-based Status-Quo Learner (SQL-PG) is,

$$f^1_{sql,pg} = \big(\alpha\cdot\nabla_{\theta^1}\mathbb{E}[R^1_0(\tau_1)] + \beta\cdot\nabla_{\theta^1}\mathbb{E}[\hat{R}^1_0(\tau_1)]\big)\cdot\delta \tag{8}$$

where $\alpha, \beta$ denote the loss scaling factor for REINFORCE, imaginative game-play respectively.

### 2.4 LEARNING POLICIES IN DYNAMIC NON-MATRIX GAMES USING $SQLoss$ AND $GameDistill$

The previous section focused on evolving optimal policies in iterated matrix games that represent sequential social dilemmas. In such games, an agent can take one of a discrete set of policies at each step. For instance, in IPD, an agent can either cooperate or defect at each step. However, in social dilemmas such as the Coin Game (Appendix A), cooperation and defection policies are composed of a sequence of state-dependent actions. To apply the Status-Quo policy gradient to these games, we present $GameDistill$, a self-supervised algorithm that reduces a dynamic game with visual input to a matrix game. $GameDistill$ takes as input game-play episodes between agents with random policies and learns oracles (or policies) that lead to distinct outcomes. $GameDistill$ (Figure 1) works as follows.

1. We initialize agents with random weights and play them against each other in the game. In these **random game-play** episodes, whenever an agent receives a reward, we store the sequence of states along with the rewards for both agents.

2. This collection of state sequences is used to train the $GameDistill$ network, which is a **self-supervised trajectory encoder**. It takes as input a sequence of states and predicts the rewards of both agents during training.

3. We then extract the embeddings from the penultimate layer of the trained $GameDistill$ network for each state sequence. Each embedding is a finite-dimensional representation of the corresponding state sequence. We **cluster these embeddings** using Agglomerative Clustering (Friedman et al., 2001). Each cluster represents a collection of state sequences that lead to a consistent outcome (with respect to rewards). For the Coin Game, when we

|     | $C$       | $D$       |
| --- | --------- | --------- |
| $C$ | (-1, -1)  | (-3, 0)   |
| $D$ | (0, -3)   | (-2, -2)  |

|     | $H$       | $T$       |
| --- | --------- | --------- |
| $H$ | (+1, -1)  | (-1, +1)  |
| $T$ | (-1, +1)  | (+1, -1)  |

|     | $C$       | $D$       |
| --- | --------- | --------- |
| $C$ | (0, 0)    | (-4, -1)  |
| $D$ | (-1, -4)  | (-3, -3)  |

(a) Prisoners' Dilemma (PD)     (b) Matching Pennies (MP)     (c) Stag Hunt (SH)

Table 1: Payoff matrices for the different games used in our experiments. $(X, Y)$ in a cell represents a reward of $X$ to the row and $Y$ to the column player. $C$, $D$, $H$, and $T$ denote the actions for the row and column players. In the iterated versions of these games, agents play against each other over several iterations. In each iteration, an agent takes an action and receives a reward based on the actions of both agents. Each matrix represents a different kind of social dilemma.

       use the number of clusters as 2, we find that one cluster consists of state sequences that represent cooperative behavior (cooperation cluster) while the other cluster represents state sequences that lead to defection (defection cluster).

4. Using the state sequences in each cluster, we **train an oracle** to predict the next action given the current state. For the Coin Game, the oracle trained on state sequences from the cooperation cluster predicts the cooperative action for a given state. Similarly, the oracle trained on the defection cluster predicts the defection action for a given state. Each agent uses $GameDistill$ independently to extract a cooperation and a defection oracle. Figure 8 (Appendix D.4) illustrates the cooperation and defection oracles extracted by the Red agent using $GameDistill$.

During game-play, an agent can consult either oracle at each step. In the Coin Game, this is equivalent to either cooperating (consulting the cooperation oracle) or defecting (consulting the defection oracle). In this way, an agent reduces a dynamic game to its matrix equivalent using $GameDistill$. We then apply the Status-Quo policy gradient to evolve optimal policies in this matrix game. For the Coin Game, this leads to agents who cooperate by only picking coins of their color (Figure 4a). It is important to note that for games such as the Coin Game, we could have also learned cooperation and defection oracles by training agents using the sum of rewards for both agents and individual reward, respectively (Lerer & Peysakhovich, 2017). However, $GameDistill$ learns these distinct policies without using hand-crafted reward functions.

Appendix B provides additional details about the architecture and pseudo-code for $GameDistill$.

## 3 EXPERIMENTAL SETUP

In order to compare our results to previous work, we use the Normalized Discounted Reward or $NDR = (1 - \gamma) \sum_{t=0}^{T} \gamma^t r_t$. A higher NDR implies that an agent obtains a higher reward in the environment. We compare our approach (Status-Quo Aware Learner or $SQLearner$) to Learning with Opponent-Learning Awareness (Lola-PG) (Foerster et al., 2018) and the Selfish Learner (SL) agents. For all experiments, we perform 20 runs and report average $NDR$, along with variance across runs. The bold line in all the figures is the mean, and the shaded region is the one standard deviation region around the mean. All of our code is available at https://github.com/user12423/MARL-with-SQLoss/.

### 3.1 ITERATED MATRIX GAME SOCIAL DILEMMAS

For our experiments with social dilemma matrix games, we use the Iterated Prisoners Dilemma (IPD) (Luce & Raiffa, 1989), Iterated Matching Pennies (IMP) (Lee & Louis, 1967), and the Iterated Stag Hunt (ISH) (Fang et al., 2002). Each matrix game in Table 1 represents a different dilemma. In the Prisoner's Dilemma, the rational policy for each agent is to defect, regardless of the other agent's policy. However, when each agent plays rationally, each is worse off. In Matching Pennies, if an agent plays predictably, it is prone to exploitation by the other agent. Therefore, the optimal policy is to randomize between $H$ and $T$, obtaining an average NDR of $0$. The Stag Hunt game represents a coordination dilemma. In the game, given that the other agent will cooperate, an agent's optimal action is to cooperate as well. However, each agent has an attractive alternative at each step, that of defecting and obtaining a guaranteed reward of $-1$. Therefore, the promise of a safer alternative

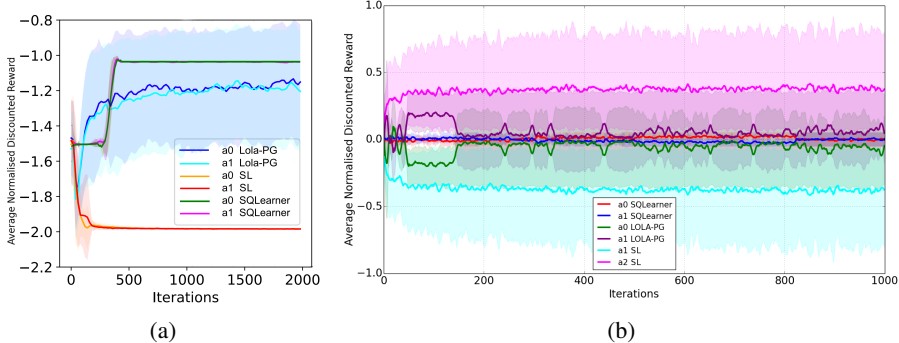

(a)                                                            (b)

Figure 3: **(a)** Average NDR values for different learners in the IPD game. $SQLearner$ agents obtain a near-optimal NDR value $(-1)$ for this game. **(b)** Average NDR values for different learners in the IMP game. $SQLearner$ agents avoid exploitation by randomising between $H$ and $T$ to obtain a near-optimal NDR value (0) for this game.

and the fear that the other agent might select the safer choice could drive an agent to select the safer alternative, thereby sacrificing the higher reward of mutual cooperation.

In iterated matrix games, at each iteration, agents take an action according to a policy and receive the rewards in Table 1. To simulate an infinitely iterated game, we let the agents play 200 iterations of the game against each other, and do not provide an agent with any information about the number of remaining iterations. In an iteration, the state for an agent is the actions played by both agents in the previous iteration.

## 3.2 ITERATED DYNAMIC GAME SOCIAL DILEMMAS

For our experiments on a social dilemma with extended actions, we use the Coin Game (Figure 5a) (Foerster et al., 2018) and the non-matrix variant of the Stag Hunt (Figure 5b). We provide details of these games in Appendix A due to space considerations.

## 4 RESULTS

### 4.1 LEARNING OPTIMAL POLICIES IN ITERATED MATRIX DILEMMAS

**Iterated Prisoner's Dilemma (IPD):** We train different learners to play the IPD game. Figure 3a shows the results. For all learners, agents initially defect and move towards an NDR of $-2.0$. This initial bias towards defection is expected, since, for agents trained with random game-play episodes, the benefits of exploitation outweigh the costs of mutual defection. For Selfish Learner (SL) agents, the bias intensifies, and the agents converge to mutually harmful selfish behavior (NDR of $-2.0$). Lola-PG agents learn to predict each other's behavior and realize that defection is more likely to lead to mutual harm. They subsequently move towards cooperation, but occasionally defect (NDR of $-1.2$). In contrast, $SQLearner$ agents quickly realize the costs of defection, indicated by the small initial dip in the NDR curves. They subsequently move towards close to 100% cooperation, with an NDR of $-1.0$. Finally, it is important to note that $SQLearner$ agents have close to zero variance, unlike other methods where the variance in NDR across runs is significant.

**Iterated Matching Pennies (IMP):** We train different learners to play the IMP game. Figure 3b shows the results. $SQLearner$ agents learn to play optimally and obtain an NDR close to 0. Interestingly, Selfish Learner and Lola-PG agents converge to an exploiter-exploited equilibrium where one agent consistently exploits the other agent. This asymmetric exploitation equilibrium is more pronounced for Selfish Learner agents than for Lola-PG agents. As before, we observe that $SQLearner$ agents have close to zero variance across runs, unlike other methods where the variance in NDR across runs is significant.

**Iterated Stag Hunt (ISH):** Appendix D.5 shows additional results for the ISH game.

## 4.2 LEARNING OPTIMAL POLICIES IN ITERATED DYNAMIC DILEMMAS

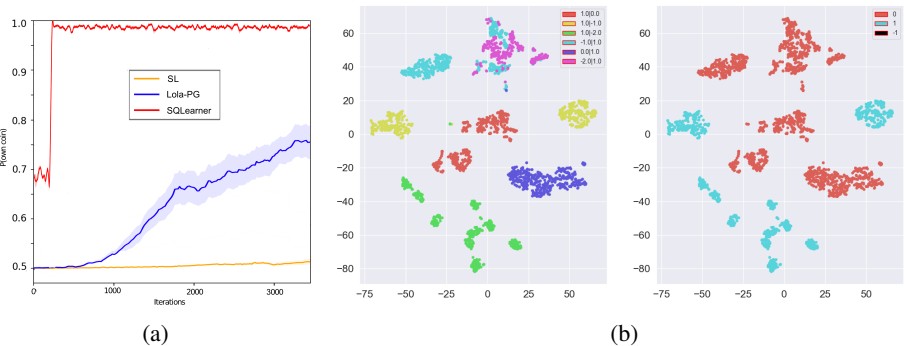

(a)                                                                              (b)

Figure 4: **(a)** Probability that an agent will pick a coin of its color in Coin Game. **(b)** Representation of clusters obtained after $GameDistill$. Each point is a t-SNE projection of the 100-dimensional feature vector output by the $GameDistill$ network for an input sequence of states. The figure on the left is colored based on rewards obtained by the Red and Blue agents. The figure on the right is colored based on clusters learned by $GameDistill$.

**GameDistill:** To evaluate the Agglomerative clustering step in $GameDistill$, we make two t-SNE (Maaten & Hinton, 2008) plots of the 100-dimensional feature vectors extracted from the penultimate layer of the trained $GameDistill$ network in Figure 4b. In the first plot, we color each point (or state sequence) by the rewards obtained by both agents in the format $r_1|r_2$. In the second, we color each point by the cluster label output by the clustering technique. $GameDistill$ correctly learns two clusters, one for state sequences that represent cooperation (Red cluster) and the other for state sequences that represent defection (Blue cluster). We experiment with different values for feature vector dimensions and obtain similar results (see Appendix B for details). Results on Stag Hunt using $GameDistill$ are presented in Appendix D.3. To evaluate the trained oracles that represent cooperation and a defection policy, we alter the Coin Game environment to contain only a single agent (the Red agent). We then play two variations of the game. In the first variation, the Red agent is forced to play the action suggested by the first oracle. In this variation, we find that the Red agent picks only $8.4\%$ of Blue coins, indicating a high cooperation rate. Therefore, the first oracle represents a cooperation policy. In the second variation, the Red agent is forced to play the action suggested by the second oracle. We find that the Red agent picks $99.4\%$ of Blue coins, indicating a high defection rate, and the second oracle represents a defection policy.

**SQ Loss:** During game-play, at each step, an agent follows either the action suggested by its cooperation oracle or the action suggested by its defection oracle. We compare approaches using the degree of cooperation between agents, measured by the probability that an agent will pick the coin of its color (Foerster et al., 2018). Figure 4a shows the results. The probability that an $SQLearner$ agent will pick the coin of its color is close to $1$. This high probability indicates that the other $SQLearner$ agent is cooperating with this agent and only picking coins of its color. In contrast, the probability that a Lola-PG agent will pick a coin of its color is close to $0.8$, indicating higher defection rates. As expected, the probability of an agent picking its own coin is the smallest for the selfish learner (SL).

## 5 CONCLUSION

We presented a status-quo policy gradient inspired by human psychology that encourages an agent to imagine the counterfactual of sticking to the status quo. We demonstrated how agents trained with $SQLoss$ evolve optimal policies in several social dilemmas without sharing rewards, gradients, or using a communication channel. To work with dynamic games, we proposed $GameDistill$, an algorithm that reduces a dynamic game with visual input to a matrix game. We combined $GameDistill$ and $SQLoss$ to demonstrate how agents evolve optimal policies in dynamic social dilemmas with visual observations.

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

*Appendix for*

STATUS-QUO POLICY GRADIENT IN MULTI-AGENT REINFORCEMENT
LEARNING

## A    DESCRIPTION OF ENVIRONMENTS USED FOR DYNAMIC SOCIAL DILEMMAS

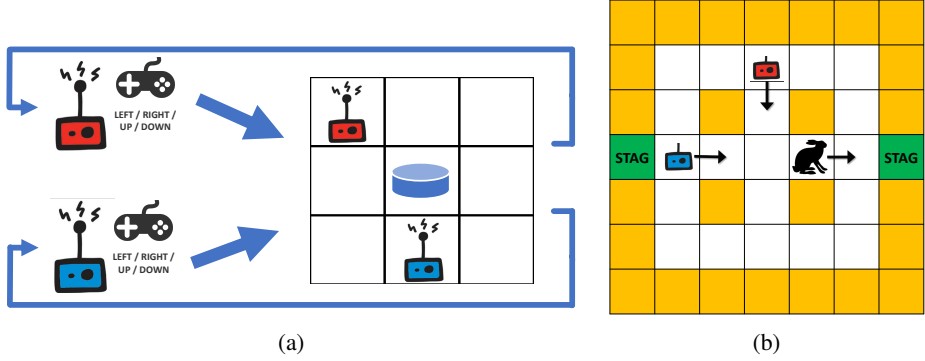

(a)                                            (b)

Figure 5: Illustration of two agents (Red and Blue) playing the dynamic games: (a) Coin Game and the (b) Stag-Hunt Game

### A.1    COIN GAME

Figure 5a illustrates the agents playing the Coin Game. The agents, along with a Blue or Red coin, appear at random positions in a $3 \times 3$ grid. An agent observes the complete $3 \times 3$ grid as input and can move either left, right, up, or down. When an agent moves into a cell with a coin, it picks the coin, and a new instance of the game begins where the agent remains at their current positions, but a Red/Blue coin randomly appears in one of the empty cells. If the Red agent picks the Red coin, it gets a reward of $+1$, and the Blue agent gets no reward. If the Red agent picks the Blue coin, it gets a reward of $+1$, and the Blue agent gets a reward of $-2$. The Blue agent's reward structure is symmetric to that of the Red agent.

### A.2    STAG-HUNT

Figure 5b shows the illustration of two agents (Red and Blue) playing the visual Stag Hunt game. The STAG represents the maximum reward the agents can achieve with HARE in the center of the figure. An agent observes the full $7 \times 7$ grid as input and can freely move across the grid in only the empty cells, denoted by white (yellow cells denote walls that restrict the movement). Each agent can either pick the STAG individually to obtain a reward of $+4$, or coordinate with the other agent to capture the HARE and obtain a better reward of $+25$.

## B    *GameDistill*: ARCHITECTURE AND PSEUDO-CODE

### B.1    *GameDistill*: ARCHITECTURE DETAILS

*GameDistill* consists of two components.

**The first component is the state sequence encoder** that takes as input a sequence of states (input size is $4 \times 4 \times 3 \times 3$, where $4 \times 3 \times 3$ is the dimension of the game state, and the first index in the state input represents the data channel where each channel encodes data from both all the different colored agents and coins) and outputs a fixed dimension feature representation. We encode each state in the sequence using a common trunk of 3 convolution layers with *relu* activations and kernel-size $3 \times 3$, followed by a fully-connected layer with $100$ neurons to obtain a finite-dimensional feature representation. This unified feature vector, called the trajectory embedding, is then given as input to

the different prediction branches of the network. We also experiment with different dimensions of this embedding and provide results in Figure 6.

The two branches, which predict the self-reward and the opponent-reward (as shown in Figure 1), independently use this trajectory embedding as input to compute appropriate output. These branches take as input the trajectory embedding and use a dense hidden layer (with 100 neurons) with linear activation to predict the output. We use the *mean-squared error (MSE)* loss for the regression tasks in the prediction branches. Linear activation allows us to cluster the trajectory embeddings using a linear clustering algorithm, such as Agglomerative Clustering (Friedman et al., 2001). In general, we can choose the number of clusters based on our desired level of granularity in differentiating outcomes. In the games considered in this paper, agents broadly have two types of policies. Therefore, we fix the number of clusters to two.

We use the *Adam* (Kingma & Ba, 2014) optimizer with learning-rate of $3e - 3$. We also experiment with K-Means clustering in addition to Agglomerative Clustering, and it also gives similar results. We provide additional results of the clusters obtained using $GameDistill$ in Appendix D.

**The second component is the oracle network** that outputs an action given a state. For each oracle network, we encode the input state using 3 convolution layers with kernel-size $2 \times 2$ and *relu* activation. To predict the action, we use 3 fully-connected layers with *relu* activation and the cross-entropy loss. We use *L2* regularization, and *Gradient Descent* with the *Adam* optimizer (learning rate $1e-3$) for all our experiments.

### B.2  $GameDistill$: PSEUDO-CODE

---
**Algorithm 1:** Pseduo-code for $GameDistill$

---
1 Collect list of episodes with $(r1, r2) > 0$ from random game play;
2 **for** *agents* **do**
3      Create dataset: $\{listEpisodes, myRewards, opponentRewards\} \leftarrow \{[\,], [\,], [\,]\}$;
4      **for** *episode in episodes* **do**
5          **for** *(s,a,r,s') in episode* **do**
6              **if** $r > 0$ **then**
7                  add sequence of last three states leading up to $s'$ to listEpisodes ;
8                  add respective rewards to myRewards and opponentRewards
9              **end**
10          **end**
11      **end**
12      Train Sequence Encoding Network;
13      Train with NetLoss;
14      Cluster embeddings using Agglomerative Clustering;
15      Map episode to clusters from Step 14;
16      Train oracle for each cluster.
17 **end**

---

### C  $SQLoss$: EVOLUTION OF COOPERATION

Equation 6 (Section 2.3.2) describes the gradient for standard policy gradient. It has two terms. The $log\pi^1(u_t^1|s_t)$ term maximises the likelihood of reproducing the training trajectories $[(s_{t-1}, u_{t-1}, r_{t-1}), (s_t, u_t, r_t), (s_{t+1}, u_{t+1}, r_{t+1}), \dots]$. The return term pulls down trajectories that have poor return. The overall effect is to reproduce trajectories that have high returns. We refer to this standard loss as $Loss$ for the following discussion.

**Lemma 1.** For agents trained with random exploration in the IPD, $Q_\pi(D|s_t) > Q_\pi(C|s_t)$ for all $s_t$.

Let $Q_\pi(a_t|s_t)$ denote the expected return of taking $a_t$ in $s_t$. Let $V_\pi(s_t)$ denote the expected return from state $s_t$.

$$
\begin{aligned}
Q_\pi(C|CC) &= 0.5[(-1) + V_\pi(CC)] + 0.5[(-3) + V_\pi(CD)] \\
Q_\pi(C|CC) &= -2 + 0.5[V_\pi(CC) + V_\pi(CD)] \\
Q_\pi(D|CC) &= -1 + 0.5[V_\pi(DC) + V_\pi(DD)] \\
Q_\pi(C|CD) &= -2 + 0.5[V_\pi(CC) + V_\pi(CD)] \\
Q_\pi(D|CD) &= -1 + 0.5[V_\pi(DC) + V_\pi(DD)] \\
Q_\pi(C|DC) &= -2 + 0.5[V_\pi(CC) + V_\pi(CD)] \\
Q_\pi(D|DC) &= -1 + 0.5[V_\pi(DC) + V_\pi(DD)] \\
Q_\pi(C|DD) &= -2 + 0.5[V_\pi(CC) + V_\pi(CD)] \\
Q_\pi(D|DD) &= -1 + 0.5[V_\pi(DC) + V_\pi(DD)]
\end{aligned}
\tag{9}
$$

Since $V_\pi(CC) = V_\pi(CD) = V_\pi(DC) = V_\pi(DD)$ for randomly playing agents, $Q_\pi(D|s_t) > Q_\pi(C|s_t)$ for all $s_t$.

**Lemma 2.** Agents trained to only maximize the expected reward in IPD will converge to mutual defection.

This lemma follows from Lemma 1. Agents initially collect trajectories from random exploration. They use these trajectories to learn a policy that optimizes for a long-term return. These learned policies always play $D$ as described in Lemma 1.

Equation 7 describes the gradient for $SQLoss$. The $log\pi^1(u_{t-1}^1|s_t)$ term maximises the likelihood of taking $u_{t-1}$ in $s_t$. The imagined episode return term pulls down trajectories that have poor imagined return.

**Lemma 3.** Agents trained on random trajectories using only $SQLoss$ oscillate between $CC$ and $DD$.

For IPD, $s_t = (u_{t-1}^1, u_{t-1}^2)$. The $SQLoss$ maximises the likelihood of taking $u_{t-1}$ in $s_t$ when the return of the imagined trajectory $\hat{R}_t(\hat{\tau}_1)$ is high.

Consider state $CC$, with $u_{t-1}^1 = C$. $\pi^1(D|CC)$ is randomly initialised. The $SQLoss$ term reduces the likelihood of $\pi^1(C|CC)$ because $\hat{R}_t(\hat{\tau}_1) < 0$. Therefore, $\pi^1(D|CC) > \pi^1(C|CC)$.

Similarly, for $CD$, the $SQLoss$ term reduces the likelihood of $\pi^1(C|CD)$. Therefore, $\pi^1(D|CD) > \pi^1(C|CD)$. For $DC$, $\hat{R}_t(\hat{\tau}_1) = 0$, therefore $\pi^1(D|DC) > \pi^1(C|DC)$. Interestingly, for $DD$, the $SQLoss$ term reduces the likelihood of $\pi^1(D|DD)$ and therefore $\pi^1(C|DD) > \pi^1(D|DD)$.

Now, if $s_t$ is $CC$ or $DD$, then $s_{t+1}$ is $DD$ or $CC$ and these states oscillate. If $s_t$ is $CD$ or $DC$, then $s_{t+1}$ is $DD$, $s_{t+2}$ is $CC$ and again $CC$ and $DD$ oscillate. This oscillation is key to the emergence of cooperation as explained in section 2.3.1.

**Lemma 4.** For agents trained using both standard loss and $SQLoss$, $\pi(C|CC) > \pi^1(D|CC)$.

For $CD$, $DC$, both the standard loss and $SQLoss$ push the policy towards $D$. For $DD$, with sufficiently high $\kappa$, the $SQLoss$ term overcomes the standard loss and pushes the agent towards $C$. For $CC$, initially, both the standard loss and $SQLoss$ push the policy towards $D$. However, as training progresses, the incidence of $CD$ and $DC$ diminish because of $SQLoss$ as described in Lemma 3. Therefore, $V_\pi(CD) \approx V_\pi(DC)$ since agents immediately move from both states to $DD$. Intuitively, agents lose the opportunity to exploit the other agent. In equation 9, with $V_\pi(CD) \approx V_\pi(DC)$, $Q_\pi(C|CC) > Q_\pi(D|CC)$ and the standard loss pushes the policy so that $\pi(C|CC) > \pi(D|CC)$. This depends on the value of $\kappa$. For very low values, the standard loss overcomes $SQLoss$ and agents defect. For very high values, $SQLoss$ overcomes standard loss, and agents oscillate between cooperation and defection. For moderate values of $\kappa$ (as shown in our experiments), the two loss terms work together so that $\pi(C|CC) > \pi(D|CC)$.

# D EXPERIMENTAL DETAILS AND ADDITIONAL RESULTS

## D.1 INFRASTRUCTURE FOR EXPERIMENTS

We performed all our experiments on an AWS instance with the following specifications. We use a 64-bit machine with Intel(R) Xeon(R) Platinum 8275CL CPU @ 3.00GHz installed with Ubuntu 16.04LTS operating system. It had a RAM of 189GB and 96 CPU cores with two threads per core. We use the TensorFlow framework for our implementation.

## D.2 SQLOSS

For our experiments with the Selfish and Status-Quo Aware Learner ($SQLearner$), we use policy gradient-based learning to train an agent with the Actor-Critic method (Sutton & Barto, 2011). Each agent is parameterized with a policy actor and critic for variance reduction in policy updates. During training, we use $\alpha = 1.0$ for the REINFORCE and $\beta = 0.5$ for the imaginative game-play. We use gradient descent with step size, $\delta = 0.005$ for the actor and $\delta = 1$ for the critic. We use a batch size of 4000 for Lola-PG (Foerster et al., 2018) and use the results from the original paper. We use a batch size of 200 for $SQLearner$ for roll-outs and an episode length of 200 for all iterated matrix games. We use a discount rate ($\gamma$) of 0.96 for the Iterated Prisoners' Dilemma, Iterated Stag Hunt, and Coin Game. For the Iterated Matching Pennies, we use $\gamma = 0.9$ to be consistent with earlier works. The high value of $\gamma$ allows for long time horizons, thereby incentivizing long-term rewards. Each agent randomly samples $\kappa$ from $\mathbb{U} \in (1, z)$ ($z = 10$, discussed in Appendix D.7) at each step.

## D.3 $GameDistill$ CLUSTERING

Figures 6 and 7 show the clusters obtained for the state sequence embedding for the Coin Game and the dynamic variant of Stag Hunt respectively. In the figures, each point is a t-SNE projection of the

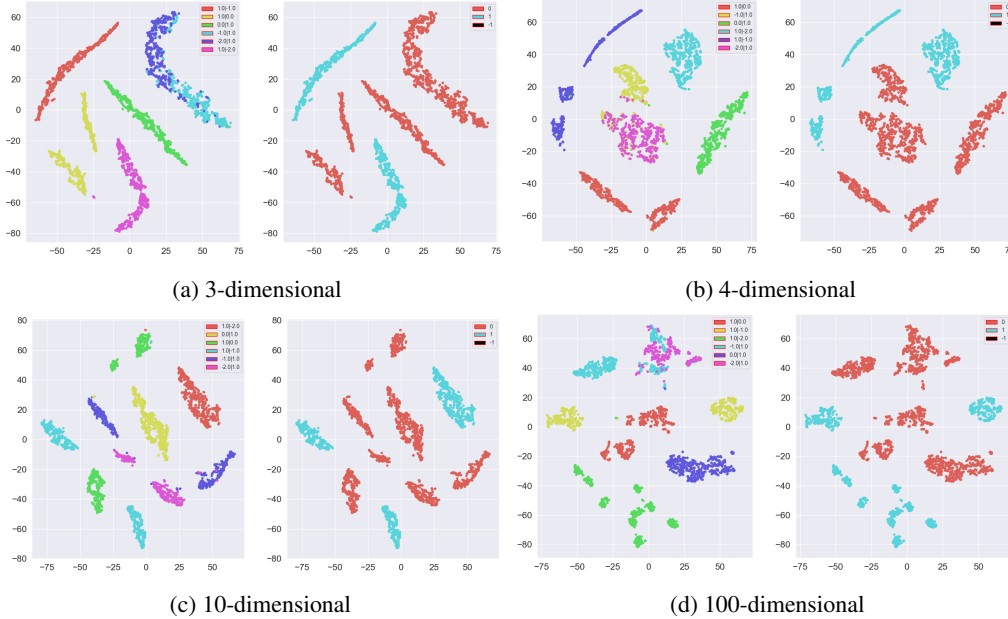

(a) 3-dimensional                    (b) 4-dimensional

(c) 10-dimensional                   (d) 100-dimensional

Figure 6: Representation of the clusters learned by $GameDistill$ for Coin Game. Each point is a t-SNE projection of the feature vector (in different dimensions) output by the $GameDistill$ network for an input sequence of states. For each of the sub-figures, the figure on the left is colored based on actual rewards obtained by each agent ($r_1|r_2$). The figure on the right is colored based on clusters as learned by $GameDistill$. $GameDistill$ correctly identifies two types of trajectories, one for cooperation and the other for defection.

feature vector (in different dimensions) output by the $GameDistill$ network for an input sequence

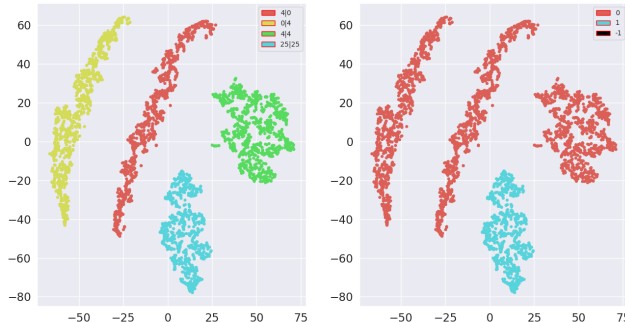

Figure 7: t-SNE plot for the trajectory embeddings obtained from the Stag Hunt game along with the identified cooperation and defection clusters.

of states. For each of the sub-figures, the figure on the left is colored based on actual rewards obtained by each agent ($r_1|r_2$). The figure on the right is colored based on clusters, as learned by $GameDistill$. $GameDistill$ correctly identifies two types of trajectories, one for cooperation and the other for defection for both the games Coin Game and Stag-Hunt.

Figure 6 also shows the clustering results for different dimensions of the state sequence embedding for the Coin Game. We observe that changing the size of the embedding does not have any effect on the results.

### D.4    ILLUSTRATIONS OF TRAINED ORACLE NETWORKS FOR THE COIN GAME

Figure 8 shows the predictions of the oracle networks learned by the Red agent using $GameDistill$ in the Coin Game. We see that the cooperation oracle suggests an action that avoids picking the coin of the other agent (the Blue coin). Analogously, the defection oracle suggests a selfish action that picks the coin of the other agent.

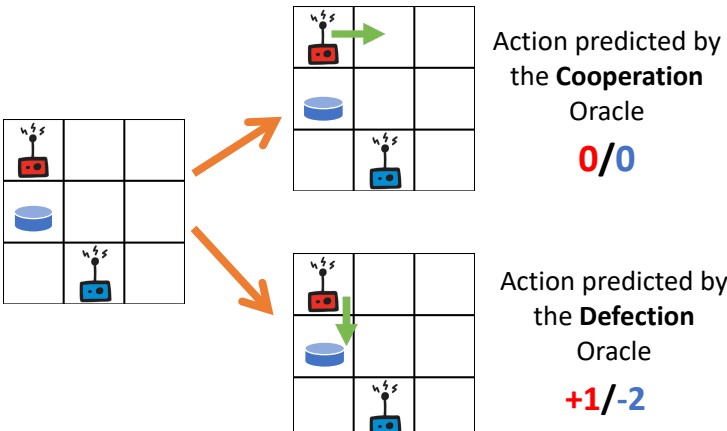

Figure 8: Illustrative predictions of the oracle networks learned by the Red agent using $GameDistill$ in the Coin Game. The numbers in red/blue show the rewards obtained by the Red and the Blue agent respectively. The cooperation oracle suggests an action that avoids picking the coin of the other agent while the defection oracle suggests an action that picks the coin of the other agent

### D.5 RESULTS FOR THE ITERATED STAG HUNT (ISH) USING SQLOSS

We provide the results of training two $SQLearner$ agents on the Iterated Stag Hunt game in Figure 9. In this game also, $SQLearner$ agents coordinate successfully to obtain a near-optimal NDR value (0) for this game.

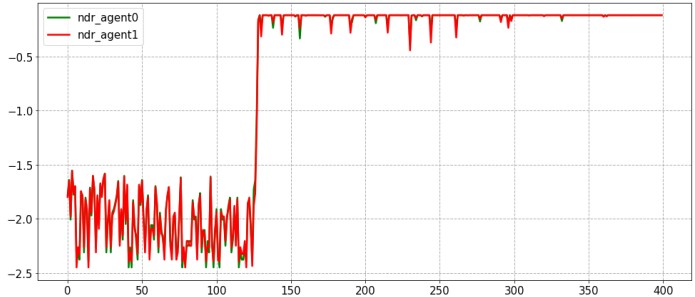

Figure 9: NDR values for $SQLearner$ agents in the ISH game. $SQLearner$ agents coordinate successfully to obtain a near optimal NDR value (0) for this game.

### D.6 RESULTS FOR THE CHICKEN GAME USING SQLOSS

We provide the results of training two $SQLearner$ agents on the Iterated Chicken game in Figure 10. The payoff matrix for the game is shown in the Table 2. From the payoff, it is clear that the agents may defect out of greed. In this game also, $SQLearner$ agents coordinate successfully to

|     | $C$        | $D$        |
| --- | ---------- | ---------- |
| $C$ | (-1, -1)   | (-3, 0)    |
| $D$ | (0, -3)    | (-4, -4)   |

Table 2: Chicken Game

obtain a near-optimal NDR value (0) for this game.

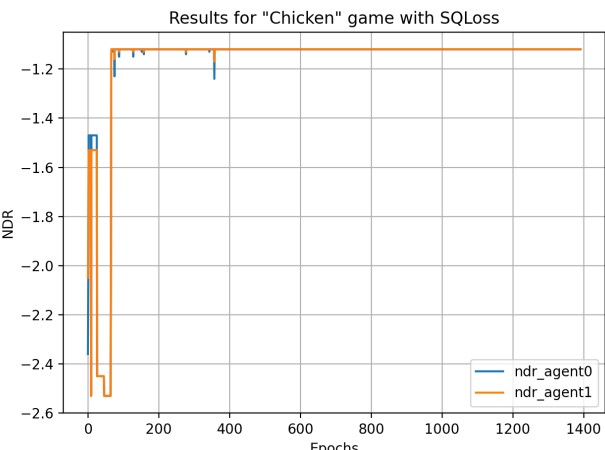

Figure 10: NDR values for $SQLearner$ agents in the Chicken game. $SQLearner$ agents coordinate successfully to obtain a near optimal NDR value ($-1.12$) for the game.

### D.7 $SQLoss$: Effect of $z$ on convergence to cooperation

We explore the effect of the hyper-parameter $z$ (Section 2) on convergence to cooperation, we also experiment with varying values of $z$. In the experiment, to imagine the consequences of maintaining the status quo, each agent samples $\kappa_t$ from the Discrete Uniform distribution $\mathbb{U}\{1, z\}$. A larger value of $z$ thus implies a larger value of $\kappa_t$ and longer imaginary episodes. We find that larger $z$ (and hence $\kappa$) leads to faster cooperation between agents in the IPD and Coin Game. This effect plateaus for $z > 10$. However varying and changing $\kappa_t$ across time also increases the variance in the gradients and thus affects the learning. We thus use $\kappa = 10$ for all our experiments.

### D.8 SQLearner: Exploitability and Adaptability

Given that an agent does not have any prior information about the other agent, it must evolve its strategy based on its opponent's strategy. To evaluate an $SQLearner$ agent's ability to avoid exploitation by a selfish agent, we train one $SQLearner$ agent against an agent that always defects in the Coin Game. We find that the $SQLearner$ agent also learns to always defect. This persistent defection is important since given that the other agent is selfish, the $SQLearner$ agent can do no better than also be selfish. To evaluate an $SQLearner$ agent's ability to exploit a cooperative agent, we train one $SQLearner$ agent with an agent that always cooperates in the Coin Game. In this case, we find that the $SQLearner$ agent learns to always defect. This persistent defection is important since given that the other agent is cooperative, the $SQLearner$ agent obtains maximum reward by behaving selfishly. Hence, the $SQLearner$ agent is both resistant to exploitation and able to exploit, depending on the other agent's strategy.

