# OpenReview forum: "Status-Quo Policy Gradient in Multi-agent Reinforcement Learning"
_ICLR.cc/2021/Conference — Reject_

### Official Review · AnonReviewer2 · 2020-10-26
**Interesting problem, but solution needs work**

**Rating:** 5
**Confidence:** 3

**Review:**

**Post-rebuttal update:** Thanks to the authors for engaging in the discussion and for the responses. The authors have provided a satisfying argument that with an appropriate choice of hyper parameters, the SQLoss does promote cooperative behaviors whenever all utilities are negative -- this addresses my main concern regarding the validity of the SQLoss objective. I remain skeptical of the value of the visual Coin game, because the results do not disentangle the usefulness of the cooperation objective and the clustering from GameDistill. I (and it seems, R1 and R3) had several concerns about the presentation of the material: mine in particular about lack of clarity, statements provided without motivation, and lack of details in Section 3, both for SQLoss and GameDistill. The authors have provided clarifications to some of these concerns in the response, but the new revision of the manuscript does not seem to reflect any of these changes. I would not be strongly opposed to acceptance, conditional on the visual coin game ablations and clarifications being added in the final version. Nevertheless, it is hard for me to recommend acceptance, given the number of unseen changes that still need to be made to the paper.




**Summary**: This paper studies the problem of learning cooperative behaviors in social dilemma problems for multi-agent deep RL with independent agents. The authors propose a simple loss that emphasizes the "status quo",  which increases the emphasis on the one-step reward, to force agents to avoid short-term exploitation. To extend this idea to a temporally-extended sequential setup, the authors introduce a clustering algorithm that reduces the game into a matrix game with "cooperate" and "defect" options.

**Assessment:** The paper studies an interesting problem: without explicit communication between two agents, how to impose protocols that ensure convergence to cooperative / optimal behaviors. The paper is also generally well-written and quite easy to understand, which I appreciated. Nonetheless, the algorithmic contributions of the paper appear lacking and incomplete. The paper does not current convince me that the proposed solution SQLoss actually solves the cooperation problem (discussion below). The GameDistill algorithm is an interesting way to reduce general environments to matrix games, but seems highly instrumented for the current environments, and unlikely to work in more general environments (discussion below). Due to my concern and confusion about the effectivity of the algorithms proposed, I currently tend to reject the paper.

**Major Concerns:**

(SQLoss) From what I ascertained from SQLoss, it changes the policy gradient to re-weigh the reward at the current time-step higher than future rewards. The text does not make it clear to me why this should force / encourage cooperative behaviors. I do understand that SQLoss will "cause an exploited agent (in DC) to ... quickly move to (DD)", but what remains confusing to me is why SQLoss will force a system in (DD) to move to (CC). If the system originally starts in a place of mutual defection (or more generally, prefers mutual defection to mutual cooperation), how does the SQLoss incentivize either player to switch to a cooperative behavior? If each agent could reason about the other agent's learning procedure, then this behavior might emerge, but the PG update does not do this: it is conditional only on the other agent's current parameters, and not how it learns. With the PG update, even when the reward at the first time-step is emphasized, neither player is incentivized to increase the probability of cooperative behavior, since cooperative behavior has strictly lower future value. One experiment that might help resolve this confuion is to show that SQLoss also induces cooperation even when the initial agents prefer mutual defection.

(GameDistill) The GameDistill algorithm is an interesting clustering method for turning a sequential game into a static game where strategies correspond to "options", but seems rather specific to the current tested environments. While the GameDistill algorithm handles discrete behaviors that "cooperate" or "defect" well, it doesn't seem to capture games where there are potentially multiple (and unknown number of) ways to cooperate or defect, or where there are graded levels of cooperation. A comment about how well GameDistill handles environment stochasticity / etc would also be helpful. Also, since it requires the practitioner to train several intermediary models (a trajectory encoder, a clustering model, an action predictor), it seems to me that the solution may be sensitive to hyperparameters, and thus not scale well to other environments. This final point is hard to judge in the current setup.

(Experiments / GameDistill) It is unclear in the experimental section, whether the performance of the SQLearner agent is due to the GameDistill algorithm reducing the dimensionality of the problem to only choosing between two options, or if it is because the SQLoss actually promotes cooperativeness. Do the other baselines, e.g. Lola-PG and SL also use GameDistill?

**Additional Comments:**

- The derivation on Page 5 is incorrect: Equation (7) does not result from $\nabla_\theta \mathbb{E}[\hat{R}_0^1(\tau_1)]$, since $\hat{R}_0^1$ contains "imagined" trajectories only for the first time-step, and not for subsequent timesteps (that is, it does not contain any terms involving $\hat{R}_t^1(\tau_1)$ for $t > 0$). It is not immediately clear to me that there is a simple objective function for which (7) is the gradient.
-  Why does the learning behavior of the SQLearner spike immediately to the optimal policy (after the initial period of being flat)?
- The related work section appears to be comprehensive, but I am not well-versed with the multi-agent literature.
- The code for running experiments is provided, which is a positive point for this paper.


**Minor Comments:**

- "Each agent independently attempts to maximize its expected discounted return": For this sentence in Section 2.1, what does the agent assume about the other agents? That the other agents are fixed? That the other agents can act adversarially?
- The word loss is used throughout for referring to both "a loss function", and a "bad reward" -- this makes the discussion in places confusing
- Use the math operator $\log$ instead of $log$

---

> ### Author Response · Authors · 2020-11-17
> **Thank you for the suggestions; Addressing the motivation behind how SQLoss leads to cooperation.**
>
> 1. We would like to thank the reviewer for the insightful observations, suggestions, and constructive feedback.
> We have tried to address the aforementioned concerns within the space limit provided.
>
>
> 2. In Appendix C, we describe the theory behind why the status quo leads to cooperative behavior.
> We summarise it here.
> Equation 6 (Section~2.3.2) describes the gradient for the standard policy gradient.
> It has two terms.
> The $log\pi^{1} (u_{t}^{1}|s_{t})$ term maximises the likelihood of reproducing the training trajectories $[(s_{t-1},u_{t-1},r_{t-1}),(s_{t},u_{t},r_{t}),(s_{t+1},u_{t+1},r_{t+1}),\dots]$.
> The return term pulls down trajectories that have poor returns.
> The overall effect is to reproduce trajectories that have high returns.
> We refer to this standard loss as $Loss$ for the following comment.
>
>
> Lemma 1:
> For agents trained with random exploration in the IPD, $Q_{\pi}(D|s_t) > Q_{\pi}(C|s_t)$ for all $s_t$.
> Please refer to Appendix C in the uploaded paper for a derivation of this Lemma.
>
> Lemma 2
> Agents trained to only maximize the expected reward in IPD will converge to mutual defection.
> This lemma follows from Lemma 1.
> Agents initially collect trajectories from random exploration.
> They use these trajectories to learn a policy that optimizes for a long-term return.
> These learned policies always play $D$ as described in Lemma 1.
>
> Equation 7 describes the gradient for $SQLoss$.
> The $log\pi^{1} (u_{t-1}^{1}|s_{t})$ term maximises the likelihood of taking $u_{t-1}$ in $s_{t}$.
> The imagined episode return term pulls down trajectories that have poor imagined return.
>
> Lemma 3:
> Agents trained on random trajectories using only $SQLoss$ oscillate between $CC$ and $DD$.
>
> For IPD, $s_{t} = (u_{t-1}^{1}, u_{t-1}^{2})$.
> The $SQLoss$ maximises the likelihood of taking $u_{t-1}$ in $s_{t}$ when the return of the imagined trajectory $\hat R_{t}(\hat \uptau_1)$ is high.
>
>
> Consider state $CC$, with $u_{t-1}^{1}=C$.
> $\pi^{1} (D|CC)$ is randomly initialised.
> The $SQLoss$ term reduces the likelihood of $\pi^{1} (C|CC)$ because $\hat R_{t}(\hat \uptau_1) < 0$.
> Therefore, $\pi^{1} (D|CC) > \pi^{1} (C|CC)$.
>
> Similarly, for $CD$, the $SQLoss$ term reduces the likelihood of $\pi^{1} (C|CD)$.
> Therefore, $\pi^{1} (D|CD) > \pi^{1} (C|CD)$.
> For $DC$, $\hat R_{t}(\hat \uptau_1) = 0$, therefore $\pi^{1} (D|DC) > \pi^{1} (C|DC)$.
> Interestingly, for $DD$, the $SQLoss$ term reduces the likelihood of $\pi^{1} (D|DD)$ and therefore $\pi^{1} (C|DD) > \pi^{1} (D|DD)$.
>
> Now, if $s_t$ is $CC$ or $DD$, then $s_{t+1}$ is $DD$ or $CC$ and these states oscillate.
> If $s_t$ is $CD$ or $DC$, then $s_{t+1}$ is $DD$, $s_{t+2}$ is $CC$ and again $CC$ and $DD$ oscillate.
> This restriction of exploration to $CC$ and $DD$ and is key to the emergence of cooperation as explained in section 2.3.1.
>
> Lemma 4
> For agents trained using both standard loss and $SQLoss$, $\pi (C|CC) > \pi^{1} (D|CC)$.
>
> For $CD$, $DC$, both the standard loss and $SQLoss$ push the policy towards $D$.
> For $DD$, with sufficiently high $\kappa$, the $SQLoss$ term overcomes the standard loss and pushes the agent towards $C$.
> For $CC$, initially, both the standard loss and $SQLoss$ push the policy towards $D$.
> However, as training progresses, the incidence of $CD$ and $DC$ diminish because of $SQLoss$ as described in Lemma 3.
> Therefore, $V_{\pi}(CD) \approx V_{\pi}(DC)$ since agents immediately move from both states to $DD$.
> Intuitively, agents lose the opportunity to exploit the other agent.
> In equation 9, with $V_{\pi}(CD) \approx V_{\pi}(DC)$, $Q_{\pi}(C|CC) > Q_{\pi}(D|CC)$ and the standard loss pushes the policy so that $\pi (C|CC) > \pi (D|CC)$.
> This depends on the value of $\kappa$.
> For very low values, the standard loss overcomes $SQLoss$ and agents defect.
> For very high values, $SQLoss$ overcomes standard loss, and agents oscillate between cooperation and defection.
> For moderate values of $\kappa$ (as shown in our experiments), the two loss terms work together so that $\pi (C|CC) > \pi (D|CC)$.
>
>
> 3. $SQLoss$ works for matrix games but is not directly applicable to games with visual input.
> GameDistill bridges this gap by reducing a stochastic game with visual input into an equivalent matrix game with a different action space consisting of cooperation and defection policies. We agree with the reviewer that in more complex games, we will need further experiments to evaluate the suitability of GameDistill.
> In our experiments so far, we have only used GameDistill for two-player games that have two strategies (cooperation and defection) and therefore exactly two clusters.
> Extending GameDistill to games with graded levels of cooperation is an area of future research.

---

> > ### Author Response · Authors · 2020-11-17
> > **Short continuation to address further points discussed by the reviewer**
> >
> > 4. To discuss the observation regarding how much of the performance of $SQLearner$ is attributable to GameDistill, we further discuss Figures 3 and 4.
> > Figure 3 shows the different methods applied to the matrix formulation of the Iterated Prisoner's Dilemma (IPD) game.
> > In this formulation, each method (SQLearner, LOLA, etc.) uses the same reduced action space consisting of cooperation and defection.
> > Figure 3 shows how agents trained using $SQLoss$ outperform those trained using LOLA in this setting.
> > For our experiments on the Coin Game (Figure 4), we use (GameDistill + $SQLoss$) against LOLA.
> > We agree with the reviewer that the current results do not conclusively disentangle the influence of GameDistill and $SQLoss$ in the Coin Game.
> > We will clarify this point in the final version of the paper.
> >
> > 5. For the derivation on page 5, we have intentionally not included the imagined actions and associated rewards in the future time-steps.
> > Our formulation is more akin to entropy addition, where the distribution over actions at a single state is modified by an additional term that need not be cumulative.
> > We believe that the formulation using imagined trajectories in future steps as well will also have a similar effect, which can be considered as a form of reward shaping.
> > In our formulation, we have the added advantage that if we were to use an Actor-Critic based approach, the Critic would have only the contribution from real-play and the imagined play will only affect the policy function of the Actor, thus displaying a bias.
> >
> > 6. We appreciate the interesting observation that the learning behavior of SQLearner spikes immediately to the optimal policy.
> > For the IPD and IMP matrix games (Figure 3), the initial number of iterations are used by agents to explore different policies, and $SQLearner$ agents converge to cooperation after the exploration phase (please refer to Appendix C for more details).
> > It is also important to state that other agents (LOLA, SL) are also relatively flat for the IPD, indicating that agents converge to stable policies once exploration has been completed.
> > For the Coin Game (Figure 4), we use GameDistill to reduce the game to its equivalent matrix formulation before training $SQLearner$ agents.
> > Therefore, $SQLearner$ agents in the Coin Game behave similarly to $SQLearner$ agents in the IPD.

---

> > > ### Comment · AnonReviewer2 · 2020-11-20
> > > **Not convinced that SQLoss is a general solution**
> > >
> > > Thank you for your response and clarifications. I have read your response, and taken a closer look again at the exact formulation in Section 2 and the discussion in Appendix C.  I have listed my remaining (and newly discovered) concerns below in order of priority. My biggest concern at the moment is that SQLoss is limited to a very small subclass of games, and will not work for even small modifications of the studied prisoner's dilemna game.
> > >
> > > **1.** Having read Appendix C, I now understand why SQLoss works on your IPD game. However, I remain unconvinced that SQLoss will work in all matrix games. For example, will SQLoss work on the following version of IPD (which just adds +3 to all the rewards)?
> > >
> > > ||C|D|
> > > |--|--------|--------|
> > > |C| (2, 2) | (0, 3) |
> > > |D| (3, 0) | (1, 1) |
> > >
> > > From my current understanding of SQLoss, it appears not. From Appendix C,
> > > - Lemma 1+2 will still be true (normal PG will converge to mutual defection)
> > > - Lemma 3 will be false. With positive reward structure, using SQLoss will not oscillate, and instead keep DD at DD, and keep CC at CC.
> > > - As a result, in your notation, $\pi(C | DD) <  \pi(D | DD)$, since *both* the PG and the SQloss will incentivize taking action D at DD.  Then, there likely is no convergence to cooperative behaviors.
> > >
> > > **2.**
> > > > the current results do not conclusively disentangle the influence of GameDistill and SQLoss in the Coin Game
> > >
> > > Given that a significant chunk of the experiments are in the Coin Game, I think it is imperative that LOLA and SL also be run with the learned options from GameDistill for a faithful comparison.
> > >
> > > **3.** The exact presentation of SQLoss in Section 2 is hard to disentangle, and needs to be made more clear. Additionally, it seems that there are a number of mathematical steps that are not justified:
> > >   - One cannot use a state-dependent baseline $b(s_t)$ for the imagined gradient (equation after Eq 7) because the gradient changes with different values of the baseline.
> > >   - My original concern about equation (7) remains unaddressed.
> > >      > For the derivation on page 5, we have intentionally not included the imagined actions and associated rewards in the future time-steps.
> > >
> > >   My statement may have been misinterpreted -- I simply meant to say that the derivation on page 5 is incorrect: (Eq 7) is just not the gradient of $\mathbb{E}[\hat{R}_0^1(\tau_1)]$.
> > >
> > > **4.** The theory in Appendix C leaves much to be desired -- it only talks about a very specific game (IPD) and many of the statements lack precision: for example, what choices of $\alpha$, $\beta$ or $\kappa$ are actually required for the lemmas to hold? This section (and the submission) would be strongly improved if these lemmas were developed for more general matrix dilemma games: perhaps the class of matrix games described by the 4 inequalities from (Leibo et al, 2017 pg 1). I doubt this is true though (point 1 above).

---

> > > > ### Author Response · Authors · 2020-11-24
> > > > **Reply regarding more general applicability of SQLoss**
> > > >
> > > > We have updated the paper with additional results.
> > > > For a given matrix game, SQLoss will work on an equivalent version of the game in which all rewards have been transformed to non-positive values.
> > > > For the matrix games described in our paper, we have used their variants with negative rewards to remain consistent with the LOLA paper.
> > > > For a general matrix game, we can subtract the maximum reward (or any number larger than it) from each reward value to make rewards negative and then use SQLoss.
> > > >
> > > > If we consider the social dilemma class of matrix games from Leibo, et. al:
> > > > $$
> > > > \begin{pmatrix}
> > > > & C & D \\\\
> > > > C & R,R & S,T \\\\
> > > > D & T,S & P,P \\\\
> > > > \end{pmatrix}$$
> > > >
> > > > where the first row corresponds to cooperation for the first player and the first column corresponds to cooperation for the second player.
> > > > Leibo, et al. define the following rules that describe different categories of social dilemmas:
> > > > 1. $R > P$ Mutual cooperation is preferred to mutual defection.
> > > > 2. $R > S$ Mutual cooperation is preferred to being exploited by a defector.
> > > > 3. $2R > T + S$ This ensures that mutual cooperation is preferred to an equal probability of unilateral cooperation and defection.
> > > > 4. either greed ($T > R$: Exploiting a cooperator is preferred over mutual cooperation) or fear ($P > S$: Mutual defection is preferred over being exploited) should hold.
> > > >
> > > > These rules have been reproduced from Leibo et. al.
> > > > 1. When "greed" ($T > R$) as well as "fear" ($P > S$) conditions hold, we have $T > R > P > S$.
> > > >     The Iterated Prisoner's Dilemma (IPD) is an example of this game.
> > > >     If we subtract $T$ from each reward, we get the matrix game
> > > > $$
> > > > \begin{pmatrix}
> > > >   & C & D \\\\
> > > > C & R-T,R-T & S-T,0 \\\\
> > > > D & 0,S & P-T,P-T \\\\
> > > > \end{pmatrix}$$
> > > > where all entries are non-positive and therefore Lemma 3 and Lemma 4 hold as before.
> > > > 2. When "greed" holds but not "fear" we have $T > R > S >= P$.
> > > >     The Chicken Game (CG) is an example of this game.
> > > >     If we subtract $T$ from each reward as before, we get the equivalent matrix game with non-positive entries and Lemmas 3 and 4 hold.
> > > > 3. When `"fear" holds but not "greed" we have $R >= T > P > S$.
> > > >     The Iterated Stag Hunt (ISH) is an example of this game.
> > > >     If we subtract $R$ from each reward, we get the equivalent matrix game with non-positive entries and Lemmas 3 and 4 hold.
> > > >
> > > > Appendix D5 and D6 in the updated paper show how agents trained with SQLoss achieve optimal returns in the Chicken and Iterated Stag Hunt games using SQLoss.
> > > >
> > > > We are running an additional experiment to disentangle the effect of GameDistill and SQLoss for the performance in Figure~4 by training a variant of LOLA that uses the oracles learned by GameDistill.
> > > >
> > > > We agree that the SQ loss term given in equation 7 is not a gradient of the expected return of the imagined trajectory in equation 5. Rather, it is a modification term that has a policy gradient-like structure. As a further consequence of this fact, the standard derivation regarding unbiased baselines does not apply here. Instead, $b(s_t)$ is a baseline-like term, though not strictly a baseline as far as the SQ loss term is concerned. We will clarify this point in the paper.

---

### Official Review · AnonReviewer1 · 2020-10-27
**Blind review**

**Rating:** 4
**Confidence:** 4

**Review:**

This paper presents a new method for improving coordination in MARL by using the idea of the status quo. The approach uses a status quo loss and a method for converting multi-step games into matrix games (called GameDistill). The methods are described and experiments are given comparing the methods to other related work.

Coordination is a problem in MARL. When multiple agents are learning at the same time, they can get stuck in poor equilibria. Therefore, ideas such as the status quo may be helpful in escaping these poor solutions.

The status quo loss is straightforward and of questionable use. The idea (in Equation 8) balances the regular RL loss witha  status quo loss that repeats the interaction (i.e., agent actions) for k steps. k is sampled from a distribution. There are weights for each of these losses to balance them out. This idea makes sense as a way to reduce the nonstationarity of decentralized learning, but it doesn't promote coordination. Why is this the right thing to do? What can be said about it theoretically? More motivation is needed for the approach.

The details of GameDistill are unclear and it also isn't clear how general it is. The method is described in text in the main paper and in pseudocode in the appendix, but each is high-level and not formal enough to understand the details. Furthermore, the approach clusters trajectories based on their rewards using random play. This seems unlikely to work well where exporation is an issue as random play may not be sufficient and in more complex games, you may need many clusters (and not know how many is needed). The paper should make it clear how general the method is.

The experiments show the method outperforms an independent learning and LOLA, but more extensive comparisons are needed. For example, the focus is on cooperative games. As such, methods that promote cooperation should also be discussed and compared to. This includes optimistic methods such as hysteresis and leniency as discussed in the paper below:

Wei, Ermo, and Sean Luke. "Lenient learning in independent-learner stochastic cooperative games." The Journal of Machine Learning Research 17.1 (2016): 2914-2955.

Also, the results are a bit surprising. The status quo loss shouldn't favor one equilibrium over another so it isn't clear why the proposed method escapes the poor equilibrium for the good one (e.g., DD for CC). The paper should make it more clear what that is the case. Lastly, the results in Figure 4 are somewhat unfair since it appears that additional training was done by the proposed method for GameDistill before learning curve plot begins.

The paper is generally well written, but some details can be more clear as mentioned above.

---

> ### Author Response · Authors · 2020-11-17
> **Thank you for the suggestions; Addressing the raised concerns below and including motivation behind why SQLoss leads to cooperation**
>
> 1. We would like to thank the reviewer for acknowledging the relevance of the problem we are trying to address in this paper. We also thank the reviewer for the insightful observations and suggestions. We have tried to address your concerns below. Please do let us know if we have misunderstood some of your observations/questions.
>
> 2. In Appendix C, we describe the theory behind why the status quo leads to cooperative behavior.
> We summarise it here.
> Equation 6 (Section~2.3.2) describes the gradient for the standard policy gradient.
> It has two terms.
> The $log\pi^{1} (u_{t}^{1}|s_{t})$ term maximises the likelihood of reproducing the training trajectories $[(s_{t-1},u_{t-1},r_{t-1}),(s_{t},u_{t},r_{t}),(s_{t+1},u_{t+1},r_{t+1}),\dots]$.
> The return term pulls down trajectories that have poor returns.
> The overall effect is to reproduce trajectories that have high returns.
> We refer to this standard loss as $Loss$ for the following comment.
>
> Lemma 1:
> For agents trained with random exploration in the IPD, $Q_{\pi}(D|s_t) > Q_{\pi}(C|s_t)$ for all $s_t$.
> Please refer to Appendix C in the uploaded paper for a derivation of this Lemma.
>
> Lemma 2:
> Agents trained to only maximize the expected reward in IPD will converge to mutual defection.
> This lemma follows from Lemma 1.
> Agents initially collect trajectories from random exploration.
> They use these trajectories to learn a policy that optimizes for a long-term return.
> These learned policies always play $D$ as described in Lemma~1.
>
> Equation 7 describes the gradient for $SQLoss$.
> The $log\pi^{1} (u_{t-1}^{1}|s_{t})$ term maximises the likelihood of taking $u_{t-1}$ in $s_{t}$.
> The imagined episode return term pulls down trajectories that have poor imagined return.
>
> Lemma 3:
> Agents trained on random trajectories using only $SQLoss$ oscillate between $CC$ and $DD$.
>
> For IPD, $s_{t} = (u_{t-1}^{1}, u_{t-1}^{2})$.
> The $SQLoss$ maximises the likelihood of taking $u_{t-1}$ in $s_{t}$ when the return of the imagined trajectory $\hat R_{t}(\hat\uptau_1)$ is high.
>
>
> Consider state $CC$, with $u_{t-1}^{1}=C$.
> $\pi^{1} (D|CC)$ is randomly initialised.
> The $SQLoss$ term reduces the likelihood of $\pi^{1} (C|CC)$ because $\hat R_{t}(\hat \uptau_1) < 0$.
> Therefore, $\pi^{1} (D|CC) > \pi^{1} (C|CC)$.
>
> Similarly, for $CD$, the $SQLoss$ term reduces the likelihood of $\pi^{1} (C|CD)$.
> Therefore, $\pi^{1} (D|CD) > \pi^{1} (C|CD)$.
> For $DC$, $\hat R_{t}(\hat \uptau_1) = 0$, therefore $\pi^{1} (D|DC) > \pi^{1} (C|DC)$.
> Interestingly, for $DD$, the $SQLoss$ term reduces the likelihood of $\pi^{1} (D|DD)$ and therefore $\pi^{1} (C|DD) > \pi^{1} (D|DD)$.
>
> Now, if $s_t$ is $CC$ or $DD$, then $s_{t+1}$ is $DD$ or $CC$ and these states oscillate.
> If $s_t$ is $CD$ or $DC$, then $s_{t+1}$ is $DD$, $s_{t+2}$ is $CC$ and again $CC$ and $DD$ oscillate.
> This restriction of exploration to $CC$ and $DD$ and is key to the emergence of cooperation, as explained in section 2.3.1.
>
> Lemma 4
> For agents trained using both standard loss and $SQLoss$, $\pi (C|CC) > \pi^{1} (D|CC)$.
> For $CD$, $DC$, both the standard loss and $SQLoss$ push the policy towards $D$.
> For $DD$, with sufficiently high $\kappa$, the $SQLoss$ term overcomes the standard loss and pushes the agent towards $C$.
> For $CC$, initially, both the standard loss and $SQLoss$ push the policy towards $D$.
> However, as training progresses, the incidence of $CD$ and $DC$ diminish because of $SQLoss$ as described in Lemma 3.
> Therefore, $V_{\pi}(CD) \approx V_{\pi}(DC)$ since agents immediately move from both states to $DD$.
> Intuitively, agents lose the opportunity to exploit the other agent.
> In equation 9, with $V_{\pi}(CD) \approx V_{\pi}(DC)$, $Q_{\pi}(C|CC) > Q_{\pi}(D|CC)$ and the standard loss pushes the policy so that $\pi (C|CC) > \pi (D|CC)$.
> This depends on the value of $\kappa$.
> For very low values, the standard loss overcomes $SQLoss$ and agents defect.
> For very high values, $SQLoss$ overcomes standard loss, and agents oscillate between cooperation and defection.
> For moderate values of $\kappa$ (as shown in our experiments), the two loss terms work together so that $\pi (C|CC) > \pi (D|CC)$.
>
> 3. We have shown how GameDistill can be used to learn cooperation and defection clusters in the Coin Game and the Stag Hunt.
> Evaluating the applicability of GameDistill to more complex games is a direction of future research.
>
> 4. We thank the reviewer for the suggestion to explore optimistic methods such as hysteresis and leniency.
> We are currently exploring these in more depth and will update our paper with a discussion comparing SQLoss to these methods.
>
> 5. In Appendix C, we show why our method escapes the poor equilibrium and converges to cooperation.
>
> 6. In Figure 4, we will add the qualification that additional training has been done using GameDistill before running the $SQLearner$ agent.

---

> > ### Comment · AnonReviewer1 · 2020-11-19
> > **Response**
> >
> > I'm still not convinced by the theory or the experiments. In terms of the theory, it is a bit difficult to follow, but it is very domain dependent. In particular, whether cooperation is encouraged by the status quo loss depends on the action set, the payoffs, the other agent policies, etc. Appendix C (and the text above) is only for one particular game, method and set of policies. What can be said more generally? And how do I balance the losses (choose $\kappa$)? I don't see how this is a general approach. GameDestill seems similarly limited, but it isn't presented in sufficient detail to understand the details. As for the experiments, since there are not comparisons with related state-of-the-art methods, it is hard to judge the benefits of the proposed method. Without a clear description of the methods along with justification and state-of-the-art comparisons, the paper cannot be accepted.

---

> > > ### Author Response · Authors · 2020-11-24
> > > **Regarding Generality of SQLoss and its applicability to different games**
> > >
> > > > I'm still not convinced by the theory or the experiments. In terms of the theory, it is a bit difficult to follow, but it is very domain dependent. In particular, whether cooperation is encouraged by the status quo loss depends on the action set, the payoffs, the other agent policies, etc. Appendix C (and the text above) is only for one particular game, method and set of policies. What can be said more generally? And how do I balance the losses (choose )? I don't see how this is a general approach.
> > >
> > > We have updated the paper with additional results.
> > > It is important to mention that agents trained with SQLoss obtain near-optimal rewards in social dilemma games with different payoff structures: IPD, IMP, ISH (both matrix and variant with visual observations), Chicken, and Coin Game (both matrix and variant with visual observations).
> > > Our method is designed to induce optimal policies in social dilemmas (as described by Leibo, et. al).
> > > For a given matrix game, SQLoss will work on an equivalent version of the game in which all rewards have been transformed to non-positive values.
> > > For the matrix games described in our paper, we have used their variants with negative rewards to remain consistent with the LOLA paper.
> > > For a general matrix game, we can subtract the maximum reward (or any number larger than it) from each reward value to make rewards negative and then use SQLoss.
> > >
> > > If we consider the social dilemma class of matrix games from Leibo, et. al:
> > > $$
> > > \begin{pmatrix}
> > > & C & D \\\\
> > > C & R,R & S,T \\\\
> > > D & T,S & P,P \\\\
> > > \end{pmatrix}$$
> > >
> > > where the first row corresponds to cooperation for the first player and the first column corresponds to cooperation for the second player.
> > > Leibo, et al. define the following rules that describe different categories of social dilemmas:
> > > 1. $R > P$ Mutual cooperation is preferred to mutual defection.
> > > 2. $R > S$ Mutual cooperation is preferred to being exploited by a defector.
> > > 3. $2R > T + S$ This ensures that mutual cooperation is preferred to an equal probability of unilateral cooperation and defection.
> > > 4. either greed ($T > R$: Exploiting a cooperator is preferred over mutual cooperation) or fear ($P > S$: Mutual defection is preferred over being exploited) should hold.
> > >
> > > These rules have been reproduced from Leibo et. al.
> > > 1. When "greed" ($T > R$) as well as "fear" ($P > S$) conditions hold, we have $T > R > P > S$.
> > >     The Iterated Prisoner's Dilemma (IPD) is an example of this game.
> > >     If we subtract $T$ from each reward, we get the matrix game
> > > $$
> > > \begin{pmatrix}
> > >   & C & D \\\\
> > > C & R-T,R-T & S-T,0 \\\\
> > > D & 0,S & P-T,P-T \\\\
> > > \end{pmatrix}$$
> > > where all entries are non-positive and therefore Lemma 3 and Lemma 4 hold as before.
> > > 2. When "greed" holds but not "fear" we have $T > R > S >= P$.
> > >     The Chicken Game (CG) is an example of this game.
> > >     If we subtract $T$ from each reward as before, we get the equivalent matrix game with non-positive entries and Lemmas 3 and 4 hold.
> > > 3. When `"fear" holds but not "greed" we have $R >= T > P > S$.
> > >     The Iterated Stag Hunt (ISH) is an example of this game.
> > >     If we subtract $R$ from each reward, we get the equivalent matrix game with non-positive entries and Lemmas 3 and 4 hold.
> > >
> > > Appendix D5 and D6 in the updated paper show how agents trained with SQLoss achieve optimal returns in the Chicken and Iterated Stag Hunt games using SQLoss.
> > >
> > > > GameDestill seems similarly limited, but it isn't presented in sufficient detail to understand the details.
> > >
> > > We have described the pseudocode for GameDistill in Appendix B2, along with a stepwise description in Section 2.4.
> > > In Appendix B1, we have described the architectural choices behind each component of GameDistill.
> > > Further, we have verified GameDistill empirically on the Coin Game and the Stag Hunt (results in Section 4 and additional plots in Appendix D3).
> > >
> > > > As for the experiments, since there are not comparisons with related state-of-the-art methods, it is hard to judge the benefits of the proposed method. Without a clear description of the methods along with justification and state-of-the-art comparisons, the paper cannot be accepted.
> > >
> > > While we have compared only with one of the most recent state of the art methods (LOLA), it is important to mention that agents trained with SQLoss obtain near-optimal rewards in the IPD, IMP, ISH (both matrix and variant with visual observations), Chicken, and Coin Game (both matrix and variant with visual observations).

---

### Official Review · AnonReviewer3 · 2020-10-28
**Effective and simple method for social dilemma matrix games; method extends to non-matrix games; motivation and explanation need improvement.**

**Rating:** 6
**Confidence:** 5

**Review:**

This paper focuses on the problem of multi-agent cooperation in social dilemmas, in which mutual defection is individually rational but collectively suboptimal. The authors use the bias toward status-quo in human psychology to motivate a new training method, called SQLoss: 1) for repeated matrix games, each agent is trained with additional imagined episodes in which the actions taken by both agents are repeated for a random number of steps; 2) for settings where cooperation and defection are associated with a sequence of actions, the authors provide a procedure called GameDistill based on trajectory encoding, clustering, and action prediction to arive at oracles for "cooperative action" and "defection action" at each state, which can then be used for the imagination episodes. Experiments show that SQL achieve better social welfare than LOLA and standard independent RL in classic iterated matrix games, as well as in the Coin Game with higher dimensional image observations.
This paper deserves a 6 mainly for showing that an agent based on a seemingly simple concept is sufficient for outperforming the well-known LOLA agent on standard benchmark matrix games. However, precisely because the method in this work seems to be so effective, there is a glaring need for improvement in many areas, and there are significant open questions for further research.

The introduction and motivation for SQLoss needs improvement. The "status quo bias" is the central concept in the paper. Hence, in the introduction, the authors must define it precisely (rather than simply relying on references, or even worse, relying on people's 'common sense' understanding of the term). The authors also should explain the main reason from human psychology that it leads to cooperation in social dilemmas. Currently, the text merely states the term and immediately explains their proposed SQLoss, saying that "SQLoss encourages an agent to stick to the action taken previously". I doubt "status-quo bias" exactly means "stick to the action taken previously". More importantly, the authors themselves say in Section 2.3.1 that "this imagined episode causes the exploited agent in (DC) to perceive a continued risk of exploitation and, therefore, quickly move to (DD)". Hence it is not correct to say that "SQLoss encourages an agent to stick to the action taken previously".

There is a key question: is there any hidden centralization at all in the method? SQLoss adds to a growing body of work that proposes methods to achieve multi-agent cooperation with decentralized training (Hughes 2018, Foerster 2018, Wang 2019, and two that the authors missed: Yang 2020 and Gemp 2020). However, there is some kind of implicit centralization in many of these works: e.g. in Hughes 2018, the centralized design of the intrinsic reward is imposed on agents prior to training; in Wang, training the reward is centralized; in Gemp 2020, all agents must "agree" before training to participate in updating the loss mixing matrix. To me, it seems that the implicit centralization required by SQLoss is in the GameDistill method, which is needed to generate the "oracles" from the temporally-extended game for use in imagination episodes. However, perhaps each agent can independently perform this preprocessing step of GameDistill, so that learning is really fully decentralized? The authors should clarify this important point.

Another key question: how does the method scale to more than two players?

Equation (4) does not seem to match the text. Section 2.3.2 says $\hat{\tau_1}$ is a collection of agent's imagined experiences. The authors say that the imagined experiences only involves the repeated actions $u^1_{t-1}$ and $u^2_{t-1}$. But in equation (4), there are additional actions $u^1_{t+1},..., u^1_T$, which is confusing.

The largest experiments in the paper (Coin Game and non-matrix Stag Hunt) are still somewhat small, even for gridworlds. Do the authors see any potential difficulties in apply the method to larger social dilemma games such as Cleanup (Hughes et al. 2018)?

Additional comments:
1) The authors use the word "evolve" in many places. This is imprecise and should be corrected, because the agents in question are learning via RL, not via evolutionary methods.
2) The introduction mentions that LOLA requires complete access to the opponent, but this is not true since LOLA has been shown to work with opponent modeling.
3) This paper used two clusters for GameDistill. Should the number always be 2?

Additional references:
Learning to Incentivize Other Learning Agents. Yang et al. 2020
D3C: Reducing the Price of Anarchy in Multi-Agent Learning. Gemp et al. 2020

---

> ### Author Response · Authors · 2020-11-17
> **Thank you for the suggestions; Addressing the raised concerns below.**
>
> 1. We would like to thank you for your insightful comments.
>
> 2. We thank the reviewer for highlighting the need for a more precise definition of the status-quo bias.
> In the paper we have tried to share the intuition by saying "SQLoss encourages an agent to stick to the action taken previously, with the \textbf{encouragement proportional} to the reward received previously."
> This proportionality implies that if the reward is highly negative, then the status-quo is relatively discouraged compared to alternatives with higher rewards.
>
> 3. We thank the reviewer for bringing up the point regarding the centralization requirements of GameDistill.
> The reviewer correctly point out that both GameDistill and SQLoss can be used independently by the agents and therefore the entire training is decentralised in nature.
> We would also like to point out the even during evaluation time we can keep learning using SQLoss because everything is decentralised.
> This makes the system adapt to non-cooperating agents during evaluation.
> We will update the paper with this point.
>
> 4. Our paper introduces a new decentralized algorithm that induces cooperation between independently learning agents, which is a challenging problem even in two-player social dilemmas.
> On the question of how our method scales to more than two players, we are currently actively exploring this area of research.
>
> 5. On the observation regarding Equation 4, we would like to point out that \phi(s_t,k_t) in Equation 3 consists of only the imagined trajectory and \hat \uptau_{1} in Equation 4 is the complete trajectory i.e. beginning with the imagined trajectory $\phi(s_t,k_t)$ and followed by real observations. Therefore the $u_{t+1},\cdots,u_{T}$ is present in the Equation 4. We will clear this up in the final version of the paper
>
> 6. We have evaluated GameDistill on two-player grid world games and have not tried games with a higher number of players.
> We believe that a significant challenge in such games will be deducing cooperation and defection strategies.
> This is because several grades of cooperation and cartels might emerge.
> This is an additional task for our future work.
>
> 7. we will change the use of the word 'evolve' and update the paper.
>
> 8. We believe that LOLA with opponent modelling would still require the design of the opponent's network architecture, observations, and learning algorithms to calculate the correct gradients.
>
> 9. We have developed SQLoss for sequential social dilemmas with cooperation and defection strategies.
> Therefore, to work with SQLoss we needed two clusters from GameDistill.
> Making SQLoss (and subsequently GameDistill) work with more than two clusters or strategies is part of our future work.

---

### Official Review · AnonReviewer4 · 2020-10-28
**simple and effective idea**

**Rating:** 7
**Confidence:** 2

**Review:**

##########################################################################

Summary:
This paper proposed a loss function that leads to cooperation between two individually optimized agents in a matrix game. The authors demonstrate this result in three matrix games. The authors further proposed a method to generalize this loss to non-matrix games and showed its effectiveness.

##########################################################################

Reasons for score:
Overall, I vote for acceptance. The idea is quite intuitive and simple while effective. The empirical result is strong and clearly demonstrates effectiveness. The authors addressed the generalization issue, which could have been a major concern, by proposing a method mapping non-matrix games to matrix games.

##########################################################################

Pros:

1.The writing is very clear and easy to follow. The paper is well-motivated and the contributions compared to previous work are clearly stated.

2. The empirical results are strong compared to the considered baselines.

3. The authors took care of the generalization issue by proposing GameDistill and presented empirical results.

##########################################################################

Cons:

1. I’m not sure what are the major differences between the three matrix games tested in the paper. It would be good to state the hypothesis and discuss what additionally we can learn from the results from different settings. Also, It seems to me that the matrix game is easily parameterized so that potentially the authors can test the loss under various parameters. Thus, though I am convinced by results from the presented three settings, it remains open whether there are bad conditions in which the loss might fail.

#########################################################################

Other questions/suggestions:
It is already quite interesting enough to only consider two agents scenario, but it seems to me that the method can be potentially extended to multi-agent (n>2) cases. It would be interesting to see a discussion or a quick experiment. I think that would make the paper even stronger.

---

> ### Author Response · Authors · 2020-11-17
> **Thank you for the feedback and suggestions**
>
> 1. We thank the reviewer for the kind words of encouragement.
>
> 2. The three matrix games tested in the paper are canonical games that appear in the sequential social dilemma literature. Hence, we selected these to demonstrate the effectiveness of SQ policy gradient approach. We have not tested our approach beyond the social dilemma setting and hence limit our claims to the same.
>
> 3. We believe that the same SQLoss formulation will scale to more than two players in inducing cooperative behavior.
> However, scaling GameDistill to more than two players is an open research problem.

---

### Decision · Program_Chairs · 2021-01-07
**Final Decision**

**Decision:**

Reject

**Comment:**

In general there is agreement under reviewers that the ideas/method presented are somewhat interesting/promising but also that the paper lacks a lot of clarity. Reviewers agree that the paper needs more work (on the method) and more extensive experiments to be convincing, and that in its current form it is not mature enough for publication at ICLR.